# Treatment of Acute Myeloid Leukemia in Older Adults

**DOI:** 10.3390/cancers15225409

**Published:** 2023-11-14

**Authors:** Aseel Alsouqi, Emily Geramita, Annie Im

**Affiliations:** Hillman Cancer Center, Division of Hematology and Oncology, University of Pittsburgh Medical Center, Pittsburgh, PA 15232, USA; alsouqia@upmc.edu (A.A.); geramitae@upmc.edu (E.G.)

**Keywords:** Acute Myeloid Leukemia, older adults, unfit adults

## Abstract

**Simple Summary:**

In this review article, we describe the landscape of available treatments for older and medically unfit patients with AML. We review the historical practice of AML treatment in older adults before discussing the current standard of care therapies and future therapeutic options. We highlight some of the challenges facing the care of older adults with AML including underrepresentation in clinical trials and shed light on the importance of evaluating clinical outcomes that are relevant to our patients and their values.

**Abstract:**

Acute Myeloid Leukemia (AML) is an aggressive myeloid malignancy predominantly affecting older adults. Despite the advancements in new therapies for AML, older and medically unfit patients continue to suffer from poor outcomes due to disease-related factors such as the mutational profile and patient-related factors such as comorbidities and performance status. In this review, we discuss a spectrum of therapeutic options for older patients with AML starting with a historical perspective and ending with therapies being investigated in clinical trials. We review the standard of care treatment options including combination venetoclax and hypomethylating agents, in addition to targeted therapies such as FLT3 and IDH inhibitors. Lastly, we shed light on challenges facing the care of older adults and their representation in clinical trials.

## 1. Background

Acute Myeloid Leukemia (AML) is predominantly a disease of older adults, with a median age at diagnosis of 68 in the United States [1]. Standard treatment of newly diagnosed AML in medically fit patients is intensive induction therapy using a combination of continuous infusion cytarabine and an anthracycline, commonly known as the “7 + 3” regimen [2,3,4]. The introduction of this regimen in the 1970s led to a drastic improvement in overall survival (OS) for patients with AML, with long-term cure rates of 30–40% in younger patients. However, in older and medically unfit patients, outcomes with intensive therapy remain dismal, with higher incidence of treatment-related toxicities across different disease risk groups, and 5-year survival rates less than 10–15% [5,6,7]. These patients are thus treated with lower intensity regimens that, historically, have been less effective than intensive regimens but spare patients the complications seen with intensive therapy.

Poor outcomes of intensive therapy in older AML patients are a result of patient-specific and disease-specific factors. Older patients are more likely to have tp53 mutated AML or secondary AML from antecedent myelodysplastic syndrome (MDS), prior exposure to alkylating agents, and/or radiation therapy, making their disease difficult to treat [8,9,10,11]. They are also more likely to have unfavorable cytogenetics and higher rates expression of multi-drug-resistant p-glycoprotein1, predisposing to resistance to chemotherapeutic agents including anthracyclines [12,13,14,15,16,17].

The past two decades witnessed an influx of effective therapies with improved toxicity profiles for older and unfit patients with AML [18,19,20,21]. In this review, we shed light on historical outcomes of lower-intensity AML treatments and then discuss current and future therapeutic options available for patients who are older or unfit for intensive chemotherapy.

## 2. Historical Perspective and Review of Agents

### 2.1. Low-Dose Cytarabine (Ara-c)

Low-dose Ara-c (LDAC) has been incorporated into the care of older AML patients since the 1970s [22,23]. LDAC exerts its effect mainly by cytotoxicity and inhibition of DNA synthesis [24,25,26]. LDAC was compared to intensive chemotherapy in a phase III randomized trial of AML patients older than age 65 [27]. The trial concluded that intensive chemotherapy resulted in higher rates of complete response (CR, 52% vs. 32%) but was associated with more frequent and severe infectious complications, increased transfusion requirements, and longer hospital stays. Both groups had similar OS rates and duration of remission. Several other reviews reported CR rates with LDAC ranging from 7% (in patients older than 70) to 32% [28,29,30]. In a meta-analysis evaluating optimal dosing of Ara-c in AML patients, LDAC was found to have no significant improvement in disease-free survival or OS in patients with unfavorable cytogenetics [30].

In current practice, LDAC monotherapy in the first-line setting has been largely replaced by combination therapies that include a hypomethylating agent and the B-cell lymphoma-2 (BCL2) inhibitor venetoclax [31]. This has been driven by the efficacy advantage these regimens offer, especially in older, higher-risk populations that cannot tolerate intensive chemotherapy. However, LDAC continues to be an option in combination with other agents, as discussed below.

### 2.2. Hypomethylating Agents

DNA methylation silences tumor suppressor genes and is an important mechanism of tumorigenesis [32]. Hypomethylating agents (HMAs) were initially used to re-induce silenced genes in the treatment of myelodysplastic syndrome (MDS). The two approved drugs for this indication, azacitidine (AZA) and decitabine (DAC), are analogues of the naturally occurring nucleoside cytidine. Early studies using HMAs proposed that the most desirable effect results from using lower doses of these drugs for longer durations to inhibit DNA methylation, rather than using higher doses, which resulted in cytotoxicity and inhibited cellular differentiation [33,34].

Both AZA and DAC were first studied in high-risk MDS patients and were shown to improve OS and the reduce risk of progression to AML when compared to best supportive care [35,36]. AZA was then compared to best supportive care, LDAC or intensive chemotherapy in a randomized phase III trial of patients with high-risk MDS [37]. Patients in the AZA arm had a significant improvement in OS compared to conventional therapies [37,38,39]. In a subgroup analysis of patients who fit the WHO AML criteria (*n* = 113, median age 70), AZA resulted in improved OS compared to the conventional therapy group [40]. This led to a phase III trial comparing AZA to a conventional care arm, which included either induction chemotherapy, LDAC, or supportive care in patients with AML who were older than 65 years [41]. Patients in the AZA arm had longer OS at 10.4 months (95% confidence interval [CI], 8.0–12.7 months) vs. 6.5 months in the supportive care group (95% CI, 5.0–8.6 months), but no survival difference was observed when AZA was compared to induction chemotherapy or LDAC. Additionally, overall response rates (ORR) and CR rates were not different among the two groups. The most common AZA related adverse events (AE) were nausea (27%), neutropenia (19%), and thrombocytopenia (17.4%). Grade 3–4 hematological-treatment-related AEs were similar across treatment groups and included febrile neutropenia (25%) and pneumonia (20.3%).

DAC was also studied in a phase III randomized trial of 485 older AML patients with high-risk features, in comparison to physicians’ choice of treatment (LDAC or best supportive care) [42]. The study showed a nonsignificant increase in median OS with DAC (7.7 months; 95% CI, 6.2–9.2) compared to physician’s choice (5.0 months; 95% CI, 4.3–6.3). There was a significant improvement of CR and CR without platelet recovery (17.8% vs. 7.8%, respectively; *p* = 0.001). Both arms had similar grade 3–4 hematological treatment-related AEs. A post hoc sensitivity analysis of the intent-to-treat population completed after data maturation demonstrated a statistically significant survival advantage at 2 years, with 14.9% (95% CI, 10.7–19.8%) in the DAC group compared to 9.9% (95% CI, 6.6–14.1%) in the control arm [43].

Overall, HMAs have efficacy in the first line for older patients with AML. Given their manageable toxicities and dosing schedule that allows for outpatient administration, HMAs are excellent partners to combine with other therapeutic classes.

### 2.3. Gemtuzumab Ozogamicin

Gemtuzumab ozogamicin (GO) is a humanized CD33 monoclonal antibody conjugated to calicheamicin that cleaves double-stranded DNA at specific sequences [44]. CD33 is a marker of normal hematopoietic progenitor cells and is expressed in over 90% of AML cells but is not expressed on hematopoietic stem cells [45].

GO was first studied at doses of 9 mg/m^2^ every 2 weeks for two doses after the first relapse [46,47,48]. Based on accumulating data of efficacy, GO was approved by the Food and Drug Administration (FDA) for treatment of relapsed AML in patients above the age of 60 in 2000. After this approval and subsequent uptake in GO use, numerous cases of vaso-occlusive disease (VOD) and sinusoidal occlusive syndrome (SOS) were reported, particularly when given at higher doses and in patients who went on to receive stem cell transplantation within 3 months of GO administration [49,50,51]. Consequently, GO was withdrawn from the United States market in 2010. GO was then studied at lower fractionated doses as a single agent in a randomized-controlled phase III trial in comparison to best supportive care [52,53]. The trial included AML patients older than 75 years or aged 61–75 with a WHO performance status of 1–2, and those who were unwilling to receive standard chemotherapy. GO was given as a single induction course of 6 mg/m^2^ on day 1 and 3 mg/m^2^ on day 8. Patients who did not progress after induction could receive up to eight monthly GO infusions of 2 mg/m^2^. The study reported a 1-year OS rate of 24.3% in the GO group compared to 9.7% in the best supportive care group. The survival benefit was largest in patients with high CD33 expression, favorable/intermediate risk disease and in women. The authors reported similar serious AEs among both groups, with no excess mortality in the GO group related to AEs.

In the relapsed/refractory setting, the Mylofrance-1 phase II study evaluated the efficacy of 3 mg/m^2^ of GO on days 1, 4 and 7 during the first relapse of AML [54]. The study showed a CR rate of 26%. At these doses, no grade 3–4 liver toxicity was seen. There were also no cases of VOD observed after GO or after stem cell transplantation in patients who received both GO and transplant. The study concluded that administration of GO in fractionated doses is likely to have a more tolerable safety profile and efficacy than initial high doses.

This series of studies suggests that higher doses of GO (9 mg/m^2^) result in high incidence of hepatic toxicity and VOD in older patients, regardless of whether GO was used as a single agent or in combination, and particularly in patients who undergo stem cell transplantation. In contrast, at fractionated doses of 6 mg/m^2^ or less, GO is effective as a single agent or in combination and results in manageable toxicities in both front-line and relapsed settings in older patients. Efficacy is enhanced in patients with favorable risk disease. Taken together, these findings have led to the use of GO as a salvage therapy in older and unfit patients with relapsed AML without targetable mutations. The benefits of efficacy with GO should be weighed against the risk of AEs including hepatic toxicity and VOD, especially in patients who may undergo stem cell transplant.

### 2.4. Venetoclax

BCL-2 is an anti-apoptotic protein that is commonly expressed in hematologic malignancies and is associated with tumorigenesis and treatment resistance [55]. Venetoclax is a selective oral BCL-2 inhibitor that has in vivo and in vitro activity against leukemic cells [56]. It was initially FDA approved in 2019 for treatment of patients with chronic lymphocytic leukemia (CLL)/small lymphocytic lymphoma (SLL). As discussed below, venetoclax is typically used in combination with HMAs or other agents due to its synergistic effects. As monotherapy, venetoclax has shown an ORR of 19% in older AML patients at a dose of 800 mg daily [57]. Patients who had isocitrate dehydrogenase (IDH) mutant disease had an even higher ORR of 33%, which has also borne out in future studies of venetoclax in combination with HMA. Venetoclax carries a risk for tumor lysis syndrome in patients with high disease burden; patients should be monitored closely either in the inpatient or outpatient setting during the initial dose escalation period [58].

### 2.5. Summary of Approach Prior to Current Era

Before the introduction of combination venetoclax and HMA, which is currently the standard of care for frontline treatment of older AML, frontline therapy for this population consisted mainly of single-agent LDAC, HMA or supportive care. While less toxic than standard induction therapy, these agents resulted in modest CR rates of 10–50%, with a median survival of around one year [37,40,41,42]. Options for relapsed/refractory disease included fractionated or low doses of GO and venetoclax, also with modest CR rates and minimal survival benefit.

## 3. Current Approach

### 3.1. Combination Venetoclax and Hypomethylating Agents

In 2020, the FDA approved venetoclax in combination with HMA for frontline treatment of AML in patients aged 75 and older and those who are unfit for induction chemotherapy, based on the VIALE-A trial [31]. Since then, this has become the standard of care for this population. This trial compared AZA alone to combination AZA and venetoclax (target dose 400 mg daily) in previously untreated older AML patients, with a 60% of the patient sample in each group older than age 75. There was a clear survival advantage for combination therapy, with a median OS of 14.7 months vs. 9.6 months in the AZA alone group (hazard ratio for death, 0.66; 95% CI, 0.52–0.85; *p* < 0.001). The combination also resulted in a higher CR rate (36.7% vs. 17.9%; *p* < 0.001). Main AEs included nausea of any grade (44% of combination group) and grade 3 or higher thrombocytopenia (45%), neutropenia (42%) and febrile neutropenia (42%). Infections of any grade occurred in 84% of the combination group and 67% of the AZA alone group, and serious AEs occurred in 83% and 73%, respectively. A recent long-term follow-up study of VIALE-A, with a median follow-up of 43 months, showed continued survival benefit for the AZA + venetoclax combination compared to AZA alone [59,60]. Emerging real-world studies provide additional evidence of benefit outside the controlled clinical trial environment. Real-world data from 230 older or unfit adults with AML who received HMA with venetoclax showed an ORR of 72% in patients receiving HMA + venetoclax and 46% in patients receiving HMA alone. The study showed a median OS of 11 months in patients receiving combination and 9 months in patients receiving HMA alone [61]. Another real world study of 112 older patients receiving venetoclax–HMA combination reported a median OS of 15 months [62].

### 3.2. Combination Venetoclax and LDAC

Venetoclax has also been studied in combination with LDAC in a phase III trial that included 211 patients randomized to receive either combination venetoclax with LDAC or LDAC with placebo [63]. The venetoclax target dose was 600 mg daily and LDAC was given at 20 mg/m^2^ subcutaneously on day 1 to day 10 of all cycles. The median age for the study population was 76, with over half of the patients in each group older than 75. Secondary AML was more frequent in the venetoclax arm. The incidence of grade 3 or higher hematological AEs including thrombocytopenia and febrile neutropenia was higher in the venetoclax group; however, this did not result in a significant difference in the rate of treatment discontinuation between groups. After a median follow up of 12 months, the median OS was 7.2 in the combination group compared to 4.1 months in the LDAC alone group, with an adjusted hazard ratio of 0.67. CR and CR with incomplete count recovery (CRi) were also significantly higher in the combination group (48% vs. 13%). In subgroup analyses of patients with somatic mutations in TP53, IDH, FMS-like tyrosine kinase 3 (FLT3) or Nucleophosmin-1 (NPM1), combination venetoclax and LDAC resulted in a CR rate of 57% in patients with IDH1 mutated AML, consistent with data seen in other studies showing increased benefit of venetoclax in IDH mutant disease. Combination venetoclax and LDAC is currently FDA approved for first line therapy in adults older than 75 years and those who are unfit to receive intensive chemotherapy. This regimen has not been compared head-to-head with the current standard of care of combination HMA + venetoclax; thus, no clear conclusions can be made regarding the superiority of either regimen. In practice, combination LDAC and venetoclax is often reserved for patients with disease progression on or after HMA or who may not be able to tolerate further HMA treatment.

### 3.3. Hedgehog Inhibitors

Glasdegib is a selective small-molecule oral inhibitor of Smoothend, a protein involved in regulating the hedgehog pathway that is implicated in CD34 + AML [64]. In a phase II trial, patients with newly diagnosed AML who were not candidates for intensive chemotherapy were randomized to receive either combination glasdegib (oral 100 mg daily) and LDAC (20 mg subcutaneously BID for 10 days in a 28-day cycle) or LDAC alone [65]. The combination demonstrated superior median OS of 8.8 months compared to 4.9 months with LDAC alone. CR rates were also higher at 17% in the combination group compared to 2.3% in patients receiving LDAC. The incidence of grade 1–2 cytopenia was higher in the combination group, though the incidence of grade 3–4 AEs was comparable between both groups. QT prolongation was observed in 5% of patients receiving glasdegib. Based on this data, glasdegib received FDA approval in 2018 to be used in combination with LDAC in previously untreated AML in older or unfit patients. This regimen has also not been compared head-to-head with venetoclax + HMA combinations; thus, the superiority of either approach cannot be concluded. Regardless, this regimen may be an option for patients who have progressed on venetoclax and/or HMA or have poor tolerance to either agent.

## 4. Targeted Therapies

### 4.1. IDH Mutant AML

Ivosidenib is a small molecule inhibitor of IDH1, which is mutated in 6–16% of patients with AML [66,67]. Early phase trials using ivosidenib in relapsed IDH1 mutant disease showed an ORR of 41.6%, with 30.4% of patients achieving CR/CRi. The median duration of response was 6.5 months. The main observed grade 3 or higher AEs were QT prolongation (7.8% of patients), differentiation syndrome (3.9%), and anemia (2.2%) [19]. Ivosidenib was then studied in the frontline setting in combination with AZA in a phase III trial that compared the combination regimen to AZA alone [68]. Ivosidenib was given orally at 500 mg daily, and AZA was given at 75 mg/m^2^ for 7 days in a 28-day cycle. At a median follow-up of 12.4 months, the study showed improved event-free survival in the combination group at 37% compared to 12% in the AZA group, with an increased OS of 24 months in the combination compared to 7.9 months in the AZA only group. While the control arm in this study was not the current standard of care regimen (venetoclax + HMA), it led to the first approval for ivosidenib in the first line setting in patients with AML. In patients with IDH1 mutated AML, ivosidenib is currently FDA approved as monotherapy for newly diagnosed and relapsed/refractory AML in patients with IDH1 mutations who are older than 75 years of age or unfit for intensive chemotherapy.

Olutasidenib is another IDH1 inhibitor that has been recently approved in relapsed/refractory IDH1 mutated AML. The pivotal trial included 147 patients with a median age of 71 and showed an ORR of 48% and a median duration of response of 11.7 months. Fifty-six-day transfusion independence was achieved in 34% of patients. Grade 3–4 treatment-related AEs included febrile neutropenia and anemia (20% each), thrombocytopenia (16%), and neutropenia (13%). Differentiation syndrome was observed in 14% of patients, with 9% being grade ≥ 3 and 1 fatal case reported [69].

Enasidenib is a small molecule inhibitor of IDH2 that is approved for use in relapsed/refractory AML. This approval was based on an early phase trial of enasidenib at a dose of 100 mg daily, which showed an ORR of 40.3% with a median response duration of 5.8 months and median OS of 19.7 months [20]. Grade 3–4 AEs included indirect hyperbilirubinemia in 12% of patients and differentiation syndrome in 7% of patients.

It is noteworthy that venetoclax has pronounced efficacy in IDH mutant AML [70]. A pooled analysis of patients enrolled in the VIALE-A phase III trial and the phase Ib trial of venetoclax + AZA in AML showed that patients harboring mutations in IDH1/2 had a higher benefit from receiving the venetoclax combination compared to AZA alone [71]. The CR rate was higher in the combination venetoclax + AZA group (IDH1 mutant: 66.7%; IDH2 mutant: 86%) compared to the AZA alone group (IDH1 mutant: 9.1%; IDH2 mutant: 11.1%). This also translated into an OS benefit of 24.5 months in the IDH1/2 mutated patients when receiving venetoclax + AZA.

Both venetoclax + HMA and ivosidenib + HMA combinations resulted in OS improvements in the first-line setting compared to HMA alone [18,68]. There have been no trials comparing either combination head-to-head; thus, both remain viable options in this setting. Future studies on the ideal sequencing of these agents in IDH-mutant AML are warranted.

### 4.2. FLT3 Mutated AML

FLT3 mutations occur in 18–31% of cases of adult AML [72,73]. Data supporting the use of FLT3 inhibitors in the first line setting in older or unfit adults are limited. Midostaurin and quizartinib are approved in combination with intensive therapy for first-line treatment of FLT3 mutated AML based on results from the RATIFY and QuANTUM trials [21,74]. In the RATIFY trial, the addition of midostaurin to intensive therapy improved overall and event-free survival; however, the trial did not include patients older than the age of 60. The QuANTUM study showed improvement in OS with the addition of quizartinib to intensive therapies and included patients up to the age of 75; however, a post hoc analysis showed that this benefit was limited to patients under the age of 60 [74,75]. Additionally, the presence of FLT3 internal tandem duplication (ITD) confers resistance to venetoclax-based therapies, which is included in most standard of care regimens for older or unfit adults [76]. Therefore, the role of FLT3 inhibitors with intensive therapies in older or unfit adults remains limited. In patients with ITD or tyrosine kinase domain mutated FLT-3, gilteritinib has been approved in the relapsed/refractory setting as monotherapy. In a phase III trial, FLT-3 mutated AML patients were randomized to receive gilteritinib orally at 120 mg daily in a 28-day cycle versus physician’s choice chemotherapy [77]. The median age for patients enrolled in the trial was 62 (range 19–85). After a median follow-up of 17.8 months, median OS was 9.3 months in the gilteritinib group compared to 5.6 months in the chemotherapy group. This improved OS was also seen in subgroup analyses evaluating patients with higher allelic ratios (>0.7 as specified by the trial) and in patients with primary refractory AML, for whom OS was 10.4 months in the gilteritinib group vs. 6.9 months in the chemotherapy group. Patients who received chemotherapy had a higher incidence of AEs in general, except for elevations in liver aminotransferase levels that was seen in the gilteritinib group. The most common treatment-related AEs were febrile neutropenia, occurring in 9.3% of patients, and QT prolongation, seen in 4.9% of patients. Gilteritinb is currently a standard of care option for relapsed FLT-3 mutated AML. Promising data are emerging to suggest the use of FLT3 inhibitors in doublets or triplets in combination with low intensity regimens as described below.

## 5. Future Directions

### 5.1. Anti-CD47 Inhibitors

CD47 is a transmembrane protein that serves as a ligand for signal regulatory protein alpha (SIRPa), which is generally found on phagocytic cells such as macrophages [78]. When activated, CD47 results in the inhibition of phagocytosis of cells and the production of a “don’t eat me signal” on tumor cells [79,80]. In studies on mouse models, CD47 was highly expressed in AML stem cells and was associated with the presence of FLT3-ITD mutations. In human AML, higher expression of CD47 was associated with poor clinical outcomes including shorter event-free and overall survival [81]. Thus, anti-CD47 antibodies have emerged as a potential therapeutic class in AML, with several early phase clinical trials reporting the safety of this class of drugs.

When used as monotherapy, CD47 inhibitors resulted in significant toxicity and low response rates in phase I trials [82]. This may be due to the low potency of those agents in increasing apoptosis, in addition to concerns for decreased selectivity to tumor cells, as CD47 are expressed on normal tissue cells as well.

To further enhance their pro-apoptotic effect, anti-CD47 antibodies were studied in combination with AZA, which upregulates the expression of calreticulin and increases phagocytosis of malignant AML cells [83]. A phase Ib trial studied combination magrolimab (5F9), a first-in-class anti-CD47 antibody, and AZA in adults with newly diagnosed AML who are unfit for intensive chemotherapy and high-risk MDS patients [84]. The study included 52 patients, 65% of whom had TP53 mutated disease. The most common treatment-related AEs were anemia (31% of patients), fatigue (19%), hyperbilirubinemia, neutropenia, thrombocytopenia, and nausea, each occurring in less than 20% of patients. Importantly, the study reported an ORR of 65%, with 44% of patients achieving CR and 12% achieving a CRi. The combination is now being studied in phase III trials (ENHANCE) in comparison to AZA and venetoclax in treatment-naïve AML and MDS (NCT04313881).

### 5.2. Menin Inhibitors

Menin is an essential protein that serves as a cofactor for KMT2A binding to HOX gene promoters, which results in leukemogenesis in patients with NPM1 or KMT2A rearranged leukemia [85,86]. Preclinical studies demonstrated the efficacy of menin inhibitors as monotherapy and in combination with other classes of agents inhibiting BCL2, CDK6 and BET inhibitors [87]. An ongoing first-in-human clinical trial is evaluating the safety and efficacy of menin inhibitor KO-539 (ziftomenib) in patients with relapsed/refractory AML (NCT04067336). SNDX-5613 (revumenib) is another potent, selective inhibitor of the menin-KMT2A interaction that is currently being studied in the phase I Augment-101 trial. A recent update on results after enrolling 66 patients with relapsed/refractory AML reported a dose limiting toxicity of grade 3 asymptomatic QTC prolongation [88]. Other AEs included grade 2 differentiation syndrome in 16% of patients, and grade 3–4 diarrhea, cytopenia, fatigue and tumor lysis syndrome. Future studies will investigate these agents in combination with standard of care for previously untreated AML with NPM1 mutations and KMT2A rearrangement in older patients.

### 5.3. Triplet Therapies

Combination therapies have advanced to the forefront of AML therapy in older adults due to the complementary mechanisms of action that combinations can achieve. The future of AML therapies will include triplet combinations that add agents to the standard of care HMA + venetoclax, with careful attention paid to minimizing toxicity.

The combination of venetoclax, cladribine (CLAD) and LDAC alternating with AZA was studied in older unfit patients in a phase II trial of 60 patients [89]. The induction regimen included intravenous CLAD on days 1–5 combined with subcutaneous LDAC on days 1–10. Venetoclax was given at a target dose of 400 mg daily. Consolidation cycles included alternating cycles of either (1) a shorter duration of CLAD (days 1–3), LDAC on days 1–10 and alternating cycles of venetoclax or (2) AZA once daily on days 1–7. The regimen achieved impressive response rates with a composite CR rate of 93% and 84% of patients achieving MRD negativity. Grade 3–4 AEs were common, with 55% of patients having febrile neutropenia and 23% developing pneumonia. After 22.1 months of follow-up, the median OS has not been reached.

In IDH1 mutant AML, triplet therapy has also showed promise. Results from a phase IB/II study of ivosidenib with AZA and venetoclax that included 25 patients with MDS and newly diagnosed or relapsed/refractory AML with IDH1 mutations showed an ORR of 92% and a composite CR rate of 84% [90]. The 1-year OS was 62%. Grade 3–4 AEs included neutropenia in 28% of patients and pneumonia in 24%.

In FLT3 mutated AML, both doublet and triplet combinations have shown promise. In a study of 54 patients with relapsed/refractory AML, combination gilteritinib and venetoclax yielded a CR rate of 74.5% at 1-year follow-up, with a median OS of 10 months [91]. A triplet combination with gilteritinib, venetoclax and AZA was studied in a phase I/II study of older and unfit patients with FLT-3 mutated AML in the frontline and relapsed/refractory setting [92]. Of the 11 previously untreated patients, the combination resulted in an ORR of 100%. The relapsed/refractory population included 15 patients, for whom ORR was 67% and median OS was 10.5 months. Given myelosuppression as the dose-limiting toxicity in 3 patients who received gilteritinib 120 mg, 80 mg was chosen as the recommended phase II dose. In a phase II trial, FLT3 inhibitors (midostaurin, gilteritinib or sorafenib) were combined with 10 days of DAC and venetoclax in newly diagnosed and previously treated AML patients. The composite CR rate was 92% in newly diagnosed patients and 62% in previously treated patients. After 14.5 months of follow-up, median OS was not reached in the newly diagnosed population and was 6.8 months in the previously treated population. The most frequent grade 3 or 4 AEs were febrile neutropenia in 40% of patients, neutropenia in 36% of patients, infections in 32% of patients and tumor lysis syndrome in 16% of patients [92]. In the frontline setting, the phase III LACEWING trial comparing gilteritinib combined with AZA to AZA monotherapy in newly diagnosed unfit patients with AML was stopped early due to futility after an interim analysis showed no significant change in the primary outcome, OS [93].

## 6. Discussion

Therapies for adults with AML who are older or unfit to receive intensive chemotherapy have evolved over the past five decades. Treatment options have become more efficacious and personalized, with little compromise in safety. We propose a personalized approach to treatment in this population that is guided by disease and patient characteristics (Figure 1). Table 1 summarizes select key studies that led to FDA approval in older or unfit adults with AML.

After initial diagnosis and determination of urgency of treatment initiation, efforts should be made to identify molecular alterations early, particularly those with therapeutic targets such as FLT3, IDH1 and IDH2. In most situations, the combination of HMA and venetoclax will be the preferred first-line treatment given robust data on safety and efficacy [18,31,59,94,95]. Cytopenias often complicate treatment with this regimen and can be managed with transfusion support, treatment breaks or even dose interruptions in venetoclax [96]. Dose reductions of venetoclax are often necessary to mitigate drug interactions with CYP3A inhibitors such as antifungal agents [97]. Additionally, patients with a high risk of tumor lysis syndrome need to be monitored closely while ramping up venetoclax. In IDH1 mutant AML, combination ivosidenib and venetoclax is an alternative first-line or subsequent-line treatment option [68]. Patients receiving IDH inhibitors should be monitored for differentiation syndrome [98]. In FLT3 mutated AML, gilteritinib is an option for relapsed/refractory disease. Patients receiving gilteritinib should be monitored for cytopenias, differentiation syndrome, hepatic toxicity and QTC prolongation [99,100]. Both combination venetoclax + LDAC and combination glasdegib + LDAC are approved in the first-line setting and may be reasonable options for patients who progress on first-line HMA + venetoclax and have no targetable mutations. GO remains an option for relapsed/refractory disease as a single agent or in combination; however, concerns have been raised regarding tolerability and hepatic toxicity. At the time of relapse, reassessing with repeat molecular studies can identify new alterations that could be targetable by available agents or clinical trials. The utilization of real-world data can provide more representative data on outcomes and toxicity in populations not reflected in the controlled clinical trial environment such as patients with low performance status or patients residing in rural areas.

Despite these improvements in available treatment options, the care of older and unfit adults with AML continues to face challenges that hinder long-term survival in this population. First, the field does not have a clear definition of “older” and “unfit” patients. Age alone is a flawed predictor of response and toxicity, as age is rarely the only factor contributing to outcomes. Select older adults with AML can be considered for allogenic hematopoietic stem cell transplant (Allo-SCT). Retrospective data showed that Allo-SCT in patients aged 60–77 was associated with longer 5-year overall survival compared to patients receiving consolidative chemotherapy despite higher initial treatment-related mortality in the group receiving allo-SCT [101]. In addition to disease and biological factors, comorbidities and the performance status play key roles in predicting the response to therapy and tolerability. In an analysis of outcomes for patients older than 70 years in the Center for International Blood and Marrow Transplant Research, the hematopoietic cell transplant specific comorbidity index, strength of the conditioning regimen, and type of donor significantly affected mortality in this population [102]. Geriatric assessment tools are more effective than provider assessment in identifying candidates for intensive therapy and predicting response and survival [103,104]. Expanded utilization of geriatric assessment tools in both clinical practice and clinical trials is needed to identify the appropriate treatment intensity and regimen for each patient. Second, older adults are generally under-represented in clinical trials due to restrictive eligibility criteria, physician attitudes about patient fitness to participate in a trial, and other social and health barriers [105]. In a survey that assessed attitudes of older adults with cancer toward clinical trial participation, the three highest ranking factors that encouraged trial participation were a recommendation from an oncologist, a chance that the patient may feel better due to the trial treatment, and a chance that the study would help other cancer patients [95]. This emphasizes the value of designing trials with meaningful clinical outcomes that focus on quality of life, as it is a key factor that patients use to evaluate treatment success. Third, both physicians and investigators need to identify clear goals of therapy when treating older and unfit patients. Despite improved outcomes with the newer regimens and agents discussed here, these approaches are not curative. As such, these therapies are given continuously until disease progression or intolerable toxicity. Providers must therefore have ongoing risk–benefit discussions with patients regarding toxicities and goals of treatment to guide treatment duration. Finally, financial toxicity due to cost of treatment, adjunctive treatments and services to manage side effects, transportation and time off work for patients and their caregivers must be considered throughout treatment to minimize the overall burden cancer patients face.

## 7. Conclusions

In conclusion, the care of older and unfit adults must be personalized. In addition to disease-specific factors that guide the selection of targeted therapies, care of these patients must be grounded in an appropriate assessment of fitness to receive therapy, functional, cognitive, and social factors, as well as patients’ goals of care. A comprehensive assessment of our patients and their needs is essential to providing the best care and improving outcomes that matter the most to our patients.

## Figures and Tables

**Figure 1 cancers-15-05409-f001:**
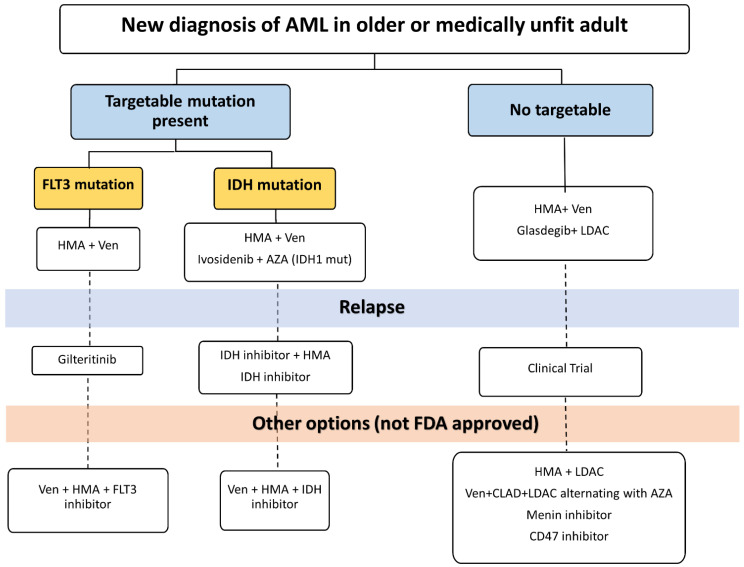
Suggested treatment algorithm for newly diagnosed AML. HMA: Hypomethylating agent. AZA: Azacitidine. LDAC: Low-dose Ara-c. CLAD: Cladribine. Ven: Venetoclax.

**Table 1 cancers-15-05409-t001:** Data from clinical trials leading to FDA approvals for current therapies for AML in older or unfit adults. AZA: Azacitidine. OS: overall survival. CR: complete response, CRi: complete response with incomplete count recovery. DAC: Decitabine. LDAC: low-dose cytarabine.

Study	Regimen	*n*	Median Age	Clinical Activity	Indication
DiNardo CD et al. [31]	Venetoclax + AZA	431	76	Median OS 14.7 months CR + CRi 66.4%	Newly diagnosed AML in patients older than 75 years, or who have comorbidities precluding induction chemotherapy
DiNardo CD et al. [18]	Venetoclax + AZA or DAC	145	74	Median OS 17.5 months CR + CRi 73%	Newly diagnosed AML in patients older than 75 years, or who have comorbidities precluding induction chemotherapy
Wei AH et al. [63]	Venetoclax + LDAC	211	76	Median OS 7.2 months CR + CRi 48%	Newly diagnosed AML in patients older than 75 years, or who have comorbidities precluding induction chemotherapy
Cortes JE et al. [65]	Glasdegib + LDAC	132	77	Median OS 8.8 months CR 17%	Newly diagnosed AML in patients older than 75 years, or who have comorbidities precluding induction chemotherapy
IDH mutated AML					
Montesinos P et al. [68]	Ivosidenib + AZA	146	76	Median OS 24 months CR 38%	First-line in IDH1-mutated AML
de Button S et al. [69]	Olutasidenib	153	71	Median OS 11.6 months	In relapsed or refractory IDH1-mutated AML
Stein EM et al. [20]	Enasidenib	239	67	Median OS 9.3 months CR 19.3%	In relapsed or refractory IDH2-mutated AML
FLT3 mutated AML					
Perl AE et al. [77]	Gilteritinib	371		Median OS 9.3 months CR + CRi 34%	In relapsed or refractory FLT3 mutated AML

## Data Availability

Data are contained within this article.

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
