# Peer review of "Treatment of Acute Myeloid Leukemia in Older Adults"

_cancers, 2023, doi:10.3390/cancers15225409_

Round 1
Reviewer 1 Report
Comments and Suggestions for Authors
In this manuscript, Alsouqi et al. aimed to discuss a spectrum of therapeutic options for older patients with acute myeloid leukemia (AML) ,their review started with a historical perspective and ended with therapies being investigated in clinical trials. They review the standard of care treatment options including combinations venetoclax and hypomethylating agents, in addition to targeted therapies such as FLT3 and IDH inhibitors. Lastly , they summarized their suggested treatment algorithm for newly diagnosed AML. Overall the review was comprehensive and interesting. However , There are also some issues that need to be addressed:
1. In this review, the enrolled patients included medically unfit old AML, the author mentioned that the treatment algorithm for newly diagnosed AML in older or medically unfit adults, including targeted therapies ,combinations venetoclax and hypomethylating agents or low dose Are-C. As we know, allogeneic hematopoietic stem cell transplantation (allo-HSCT) is also important treatment option for older patients, I suggest the author to discuss the role of allo-HSCT in older patients.
2. Various inhibitors of FLT3 are widely used in elderly AML with large role , and the authors are invited to elaborate in this section.
3. Elderly AML is not always Unfit; patients who are Unfit at the time of initial diagnosis can change to Fit patients after obtaining a complete remission, and the issue of post-treatment re-evaluation should also be emphasized when presenting all-HSCT for the treatment of elderly AML.
Author Response
1. Thank you for the important comment. We agree that allo-HSCT is an important treatment option for older patients with AML, and we added brief text in this regard. Our review focuses on unfit patients, so we left this part brief.
2. Thank you for this comment, we expanded on this section.
3. Thank for this comment, we agree and added text in regards to this.
Reviewer 2 Report
Comments and Suggestions for Authors
In this review, the authors shed light on historical outcomes of lower intensity AML treatments and then discussed current and future therapeutic options available for patients who are older or unfit for intensive chemotherapy. This is a comprehensive and exhaustive review that reported several clinical trials. The review can be accepted with minor revision
However, in my opinion, to make the review more interesting and different from others some real word experience of the different drugs should be reported at least in the discussion. Among the studies to be included is that of Candoni A. Am J Hematol. 2023 Apr;98(4):E80-E83. doi: 10.1002/ajh.26846. Epub 2023 Feb 9.
Author Response
Thank you for this important comment, we expanded on this topic and added the reference as noted.